# Complementary and Integrative Therapies for Childhood Atopic Dermatitis

**DOI:** 10.3390/children6110121

**Published:** 2019-10-30

**Authors:** Adrienne L. Adler-Neal, Abigail Cline, Travis Frantz, Lindsay Strowd, Steven R. Feldman, Sarah Taylor

**Affiliations:** 1Center for Dermatology Research, Department of Dermatology, Wake Forest School of Medicine, Winston-Salem, NC 27101, USA; aecline25@gmail.com (A.C.); tfrantz@wakehealth.edu (T.F.); sfeldman@wakehealth.edu (S.R.F.); 2Department of Dermatology, Wake Forest School of Medicine, Winston-Salem, NC 27101, USA; lchaney@wakehealth.edu (L.S.); saratayl@wakehealth.edu (S.T.); 3Department of Pathology, Wake Forest School of Medicine, Winston-Salem, NC 27101, USA; 4Department of Public Health Sciences, Wake Forest School of Medicine, Winston-Salem, NC 27101, USA

**Keywords:** integrative medicine, atopic dermatitis, children

## Abstract

Background: Childhood atopic dermatitis is a chronic inflammatory skin condition that causes significant psychological and financial costs to the individual and society. Treatment regimens may require long-term medication adherence and can be associated with poor patient satisfaction. There is considerable interest in complementary and integrative medicine (CIM) approaches for childhood atopic dermatitis. Objective: To assess the effects of CIM approaches on childhood atopic dermatitis outcomes as defined by randomized, controlled clinical trials. Methods: A PubMed review of CIM-related treatments for pediatric atopic dermatitis was performed, and data related to age, study population, efficacy, treatment regimen, length of treatment, and sample size were included. Results: The search yielded 20 trials related to probiotic/prebiotic treatments for atopic dermatitis, three on the effects of vitamins on children with atopic dermatitis, and two on the effects of Chinese herbal treatments for atopic dermatitis in children and adolescents. The strongest evidence was for supplementation with the probiotics *L. fermentum* and *L. plantarum*. Conclusions: Certain strains of probiotics, specifically *L. plantarum* and *L. fermentum,* may improve clinical severity scores in children with atopic dermatitis. However, additional trials are needed to more thoroughly delineate the effects of additional integrative therapies on childhood atopic dermatitis.

## 1. Introduction

Atopic dermatitis (AD) is the most common inflammatory skin condition seen in pediatric populations. Its prevalence in the past twenty years has been steadily increasing, reaching a rate of 13% in the United States [1,2]. A complex constellation of risk factors including irritants, contact and inhaled allergens, stress, and infection contribute to the development and persistence of AD [3]. Clinically, AD is characterized by itching, skin inflammation, skin barrier abnormalities, and increased susceptibility to skin infections [4]. This condition negatively impacts quality of life for affected children and their caregivers and can lead to significant decreases in self-esteem, as well as increased rates of depression, anxiety, and suicidal ideation [5,6,7,8,9].

Despite the existence of effective treatment regimens, poor patient satisfaction with the treatment is common. Less than one third of patients report satisfaction with their current treatment regimens [10]. There is interest in complementary and integrative medicine (CIM) for the treatment of childhood AD, with over 40% of pediatric patients with AD reporting the use of integrative medicine approaches [11]. Recommendations for CIM approaches should be based on evidence. We assessed the effects of CIM on childhood atopic dermatitis outcomes as defined by randomized, controlled clinical trials (RCTs).

## 2. Methods

A PubMed search included the terms “atopic dermatitis,” “children,” “pediatrics,” “integrative medicine,” “complementary medicine,” “alternative medicine,” “complementary,” “integrative,” “alternative,” “probiotic,” “herb,” “herbal medicine,” “vitamin,” and “relaxation”. Results were filtered to only include human subjects. Dates included ranged from the earliest entry to January of 2019. Title, abstract, and full review were conducted by one reviewer. We included only randomized controlled trials (RCT) or crossover trials that examined participants less than or equal to eighteen years of age. An exception was made for one study examining Chinese herbal medicine that incorporated individuals 5–21 years of age.

When using the above search terms, anywhere between approximately twenty to eighty trials resulted related to probiotics for the treatment of AD. From those, ten were identified as meeting our study criteria, while the remaining ones cited below were found within the References section of additional articles. For the section related to vitamin supplementation, approximately ninety-five studies resulted from our search, with only one meeting our criteria. An additional two studies were also found from the References section of articles. For studies related to herbal medicine, approximately forty studies were identified on PubMed, two of which fit our classifications.

## 3. Results

A total of 20 RCTs examined the role of probiotics/prebiotics in the treatment of childhood AD [12,13,14,15,16,17,18,19,20,21,22,23,24,25,26,27,28,29,30,31]. Three RCTs examined the effects of vitamins on children with AD [32,33,34], and two RCTs delineated the effects of Chinese herbal treatments for AD in children and adolescents (Table 1) [35,36].

### 3.1. Probiotics/Prebiotics for the Treatment of Childhood Atopic Dermatitis 

Of the 20 RCTs that examined the role of a variety of probiotic/prebiotic mixtures on AD, the most common probiotics used were *L. acidophilus, L. rhamnosus,* and *L. plantarum* (Table 1) [12,13,14,15,16,17,18,19,20,21,22,23,24,25,26,27,28,29,30,31]. A wide array of other probiotics for AD were used (Table 2). The majority of the treatment durations in these studies were between 6 and 12 weeks.

Of these, three trials examined children under one year of age [23,25,27]. None of the three trials solely examining children under one year of age showed any improvement in disease severity when compared to placebo. However, a small number of specific probiotic strains were incorporated in these three studies (i.e., *L. paracasei*, *B. lactis*, *L. rhamnosus*, *B. breve*, *P. freudenreichii*, and *Shermanii JS*), thus limiting the extension of these findings to all probiotic treatments.

Four studies examined *L. acidophilus* (both delivered on its own and in combination with additional probiotics) for the treatment of AD [17,19,21,24]. One RCT of 50 children ages 4–15 compared *L. acidophilus* to placebo treatment [24]. The primary outcome measure was the symptom-medication score (SMS). This score combined the AD Area and Severity Index (ADASI) with the medication score, a marker that represented the amount of steroid ointment used. The treatment length was eight weeks, after which the probiotic group showed greater improvements in the SMS when compared to placebo (*p* < 0.05). No specific information was provided in this article concerning the amount of improvement observed in either group [24]. In an additional RCT, *L. acidophilus* was combined with multiple additional probiotic strains as well as a prebiotic (i.e., synbiotic) and compared with placebo treatment for a duration of 8 weeks in 40 children ages 3 months–6 years [17]. After 8 weeks of treatment, those assigned to the synbiotic group demonstrated a greater decrease in SCORAD index (−39 points) when compared to placebo treatment (−20 points, *p* = 0.005) [17]. *L. acidophilus* treatment also improved the SCORAD index more than placebo did in a separate RCT that treated 90 children ages 12–36 months for 8 weeks with a combination of *L. acidophilus* and *B. lactis* [19]. The treatment group experienced a 14 point reduction in the SCORAD index when compared to an 8 point improvement in the placebo group (*p* = 0.001) [19]. The combination of *L. acidophilus* and *B. lactis* was also associated with decreased CD4 and CD25 values, a factor that correlated with improvements in the SCORAD index (*p* < 0.05) [19]. Further, when *L. acidophilus* was combined with *B. bifidum*, *L. casei*, and *L. salivarius*, the treatment group exhibited a 65% reduction in the SCORAD index when compared to the 46% reduction observed in the placebo group (*p* = 0.002) [21]. This combination of *L. acidophilus*, *B. bifidum*, *L. casei*, and *L. salivarius* also improved inflammatory markers such as IL-6, IFN-γ, and IgE when compared to placebo (*p* < 0.01) [21].

Three RCTs examined the role of *L. plantarum* in treating childhood AD [14,16,22]. In one trial, 12 weeks of treatment with *L. plantarum* in children ages 0–14 improved an average of 37 points in the probiotic group as compared to 27 points in the placebo group (*p* < 0.001) [14]. In an additional RCT, 12 weeks of treatment with *L. plantarum* in children ages 1-13 was associated with a 9-point decrease in the SCORAD index as compared to a 2 point decrease in the placebo group, a group difference that was statistically significant (*p* = 0.004) [22]. Interleukin (IL)-4, interferon (IFN)-γ, and IL-17 were also significantly lower in the probiotic group when compared to placebo. A separate trial combined *L. plantarum* with *L. casei*, *L. rhamnosus*, and *B. lactis* and the effects of six weeks of this regimen were compared with placebo treatment in children 2–9 years old [16]. The probiotic group showed a 35% decrease in Eczema Activity and Severity Index (EASI) scores, whereas the placebo group demonstrated a 46% decrease in EASI scores. There were no significant group differences in changes in EASI scores (*p* = 0.28) [16].

Two RCTs examined *L. fermentum* as a treatment for pediatric AD [18,31]. In the first RCT, 16 weeks of treatment with *L. fermentum* was compared with placebo in 53 children ages 6–18 months old [18]. Probiotic treatment improved SCORAD values an average of 17 points (*p* = 0.03), whereas placebo treatment did not result in statistically significant improvements (−12 points, *p* = 0.83). No statistical data were provided regarding intergroup comparisons of these measures [18]. An additional trial treated children ages 1-18 years with either *L. paracasei*, *L. fermentum*, *L. paracasei* + *L. fermentum*, or placebo for 12 weeks, and the effects on SCORAD index and inflammatory markers (IL-4, IFN-γ, transforming growth factor (TGF)-ß, and tumor necrosis factor (TNF)-α) were measured [31]. SCORAD index improved by 25 points in those treated with *L. paracasei,* 24 points in those treated with *L. fermentum*, 28 points in those given *L. paracasei* + *L. fermentum*, and 15 points in those treated with placebo. All groups demonstrated lower SCORAD indices when compared to placebo and after adjusting for topical steroid use (*p* < 0.001) [31]. There were no group differences in topical steroid use. Treatment with *L. fermentum* and *L. paracasei* also decreased IL-4 levels (*p* = 0.04) [31]. 

One RCT combined *L. salivarius* with a prebiotic (i.e., synbiotic) and compared it with the use of a prebiotic alone in 54 children ages 2–14 years [28]. Treatment was administered for 8 weeks. At week 8, the synbiotic group demonstrated an average SCORAD index of 27, whereas the placebo group had an average SCORAD index of 36. Significant group differences were observed when controlling for baseline SCORAD values (*p* = 0.02) [20]. One RCT examined *L. sakei* treatment in comparison to placebo in 75 children ages 2–10 years old [29]. The probiotic group demonstrated improvements in SCORAD values (−31%) that were greater than those observed in the placebo group (−13%, *p* = 0.01). The treatment group also demonstrated improvements in cytokine levels, a factor that was associated with improvements in clinical scores (*p* < 0.001) [29]. 

While four trials incorporated *L. casei* into the probiotic mixture, no RCTs have examined *L. casei* alone. As noted above, *L. casei* was ineffective when combined with *L. plantarum, L. rhamnosus,* and *B. lactis* (*p* = 0.28) [16]. However, *L. casei* was effective when combined with additional strains that may deliver more potent treatment effects (also noted above, i.e., *L. plantarum*, *L. acidophilus*, and *L. salivarius*) (*p* < 0.005) [17,21]. When *L. casei* was combined with *B. lactis* and *B. longum* and examined in 47 individuals 4–17 years of age for 12 weeks, significant improvement in SCORAD values were observed in the treatment group (−83%) when compared to placebo (−24%, *p* < 0.001) [13]. *B. lactis* was also effective when combined with *L. acidophilus* (SCORAD improved by 14 points, *p* = 0.001) and was ineffective when used on its own or when combined with additional strains [12,16,19,23].

Eight trials incorporated the *L. rhamnosus* strain [12,15,16,17,20,25,26,27]. Of those, three demonstrated improvements in treatment outcomes when compared to placebo treatment [20]. Specifically, one RCT compared *L. rhamnosus*, *L. rhamnosus* + multiple additional strains, and placebo treatment in 208 children ages 1.5–11.9 months [25]. No group differences were found in the SCORAD index after four weeks of treatment (*p* = 0.27). In an additional trial, 102 children 3–12 months of age were administered *L. rhamonsus* or placebo for 12 weeks and changes in the SCORAD index were measured. Again, no group differences were seen [27]. However, it should be noted that in both of these trials, all of the participants were all under one year of age. In the trial by Yang et al. (discussed above), a probiotic mixture incorporating *L. rhamnosus* demonstrated no efficacy when compared to placebo (*p* = 0.28). In two RCTs, children between the ages of one month and 12 years experienced no benefits of *L. rhamnosus* when compared to a prebiotic treatment alone [15] and when compared to placebo [26]. However, one RCT showed that eight weeks of *L. rhamnosus* treatment in 62 children 4–48 months of age led to improvements in the SCORAD index (−22 points) when compared to placebo (−12 points, *p* = 0.01) [20]. Additionally, L. rhamnosus was combined with B. lactis and showed improvement in children who were food sensitized when compared to placebo (SCORAD ratio of probiotic to placebo treatment was 0.73 (*p* = 0.047) [12]. The strain was also effective when combined with a number of different probiotics and a prebiotic (−39 points) when compared to placebo (−20 points, *p* = 0.005) [17].

Three RCTs listed above included prebiotics with probiotics (i.e., synbiotics) [15,17,28]. Synbiotics had a greater impact on the SCORAD index (39 point improvement) than prebiotics alone (20 point improvement) when the probiotics used was *L. salivarius* (*p* = 0.005) [17]. Treatment with a synbiotic was also more effective than a placebo when the probiotics used were a mixture combining *L. casei*, *L. rhamnosus*, *S. thermophilus*, *B. breve*, *L. acidophilus*, *B. infantis*, and *L. bulgaricus* [17]. Prebiotics alone are also effective in improving the SCORAD index (−39%, *p* < 0.05) [15]. In one study not discussed above, 12 weeks of treatment with a synbiotic in 37 children ages 1–36 months old had no effect on SCORAD index when compared to placebo (*p* > 0.05) [30]. However, the probiotic strains used were not listed.

All trials supported the safety of probiotic treatments in children. Few adverse events were observed, and the ones that occurred were mild. One participant reported vomiting after the ingestion of the probiotic [18], while three individuals in another study reported mild abdominal pain after ingestion of a prebiotic and synbiotic [15]. Seventeen participants in an additional study reported side effects such as vomiting, diarrhea, and fever after ingestion of active treatment. However, these side effects were seen in the placebo group as well and no significant group differences were observed [26]. 

### 3.2. Vitamins/Minerals for the Treatment of Childhood Atopic Dermatitis 

Oral pyridoxine, oral zinc sulfate, and topical B12 treatments have been examined in randomized, placebo and vehicle-controlled trials for the treatment of childhood AD (Table 3) [32,33,34]. Twenty-one patients ages 6 months-18 years treated with topical B12 ointment for 4 weeks demonstrated improvement in SCORAD values at 2 weeks (−3 points) when compared to vehicle (−1 point, *p* = 0.01) and 4 weeks (−4.5 points) when compared to the vehicle (−1.6 points, *p* = 0.01) [33]. Four weeks of treatment with oral pyridoxine in children ages 2–15 and eight weeks of treatment with zinc sulfate in those 1–16 years old showed no effects when compared to placebo/vehicle treatment [32,34]. No significant adverse events were noted as a result of vitamin/mineral ingestion.

### 3.3. Chinese Herbs for the Treatment of Childhood Atopic Dermatitis

Two RCTs examined the role of traditional Chinese herbal medicine (TCHM) in improving the severity of childhood AD (Table 4) [35,36]. One trial incorporated the following herbs into the TCHM mixture: *Flos Ionicerae, Herba menthae, Cortex moutan, Rhizoma atractylodis, and Cortex phellodendri* [35]. Twelve weeks of treatment was compared to placebo in 85 individuals between the ages of 5 and 21 years. TCHM resulted in an improvement of 15% in SCORAD index, while placebo treatment caused an improvement of 18%. There were no group differences in changes in SCORAD values. However, treatment with TCHM did result in a reduction in the total amount of steroids used (*p* = 0.02). A different, crossover trial examined the effects of a TCHM treatment on clinical severity scores in 37 children ages 1–18 years of age [36]. The TCHM mixture incorporates *Ledebouriella seseloides, Potentilla chinensis, Anebia clematidis, Rehmannia glutinosa, Paeonia lactiflora, Lophatherum gracile, Dictamnus dasycarpus, Tribulus terrestris, Glycyrrhiza uralensis, and Schizonepeta tenuifolia*. TCHM treatment was associated with a 51% decrease in erythema scores as compared to a 6.1% decrease during placebo treatment (95% CI for the difference 13.4, 89.7). TCHM treatment was also associated with a 49% decrease in surface damage scores when compared to a 6.2% decrease during placebo treatment (95% CI for the difference 19.2, 97.9) [36]. No adverse events were noted due to TCHM ingestion.

## 4. Discussion

The treatment of pediatric AD can be difficult and may benefit from the combination of allopathic and integrative treatment approaches. To date, probiotics remain the most well-validated CIM approach for treating childhood AD. Appropriate development of the immune system, including the maturation of Th1/Th2 cells in childhood, depends on adequate microbial stimulation [37]. In fact, microbial composition and diversity prevent the shift towards Th2-mediated immunity and the subsequent development of allergic diseases [2,38]. As such, probiotic treatments may improve AD clinical severity through their immunomodulatory effects [29,31]. However, the dosage of probiotic treatments was not standardized across studies and is thus a limitation of the data presented. The strongest evidence exists for treatment with *L. plantarum* and *L. fermentum* in children 12 months of age and older. Specifically, each probiotic was examined separately in two RCTs that demonstrated a reduction in the SCORAD index when *L. plantarum* or *L. fermentum* was delivered alone (i.e., without additional probiotic strains) [14,18,22,31]. Many of the improvements seen were clinically significant since on average, an improvement of 8.7 points on the SCORAD index equates to a one-point improvement on the global severity scale [39]. While one trial did not show any efficacy of *L. plantarum* treatment [16], this study only incorporated six weeks of treatment, while the former RCT’s delivered 12 weeks of treatment, potentially highlighting an effect of longer treatment times. This trial also combined *L. plantarum* with potentially less effective probiotic strains (i.e., *L. casei* and *B. lactis*) [16].

Treatment with *L. acidophilus*, *L. salivarius*, *L. paracasei*, and *L. sakei* may also provide benefit when delivered without additional strains. However, each of these bacteria alone has been evaluated in only one RCT [24,28,29,31]. *L. acidophilus* was also effective in improving the SCORAD index when combined with additional strains [19,21]. *L. rhamnosus* may be effective for improving AD-related outcomes [12,15,16,17,20,25,26,27]. Multiple studies examining this strain were performed in children less than one year of age, a factor that could have contributed to the lack of efficacy. Further, while three studies showed some improvement, only one of these examined *L. rhamnosus* alone [20]. Future studies are needed to examine the efficacy of *B. lactis* and *L. casei*, as conflicting findings were noted in numerous studies [12,13,16,19,23]. Preliminary evidence suggests that synbiotics may improve AD-related outcomes but demonstrate additional efficacy when combined with an active probiotic [15,17,28]. Limited evidence exists supporting the efficacy of topical B12 ointment [33]. No conclusive evidence exists for the use of TCHM in children with AD [35,36].

Notably, few RCTs have examined the role of mind-body approaches in treating childhood AD. Psychological stress is a key component of an individual’s experience with AD and can even exacerbate the disease processes [40]. Interventions that mitigate stress, such as progressive muscle relaxation, mindfulness meditation, and yoga could provide a cost-effective treatment approach in this patient population. However, it remains unknown if mind-body therapies confer benefit in pediatric populations suffering from AD. Outside of probiotic therapies, little to no evidence exists regarding the efficacy of CIM treatments for pediatric AD.

In summary, while CIM approaches hold promise for treating childhood AD and so far are not known to hold many side effects, it remains unknown how these approaches will fit into the broader landscape of traditional AD treatments. Consequently, future clinical trials are needed to comprehensively understand the role of herbal remedies, vitamins, cognitive therapies, relaxation techniques, and additional CIM modalities in treating pediatric AD.

## Figures and Tables

**Table 1 children-06-00121-t001:** Probiotics/prebiotics for the treatment of pediatric atopic dermatitis.

Study	Study Type	Treated/Control (N)	Age Range	Probiotic/Prebiotic Treatment and Daily Dosage	Treatment Length	Outcome Measure	Results
Viljanen et al., 2005	RCT	75/67/66	1.5–11.9 months	*L. rhamnosus GG* (*LGG*) 10^10^ cfu vs. *LGG* 10^10^ cfu, *L. rhamnosus* (*LC705*) 10^10^ cfu, *B. breve* 4 × 10^8^ cfu, *Propionibacterium freudenreichii*, *Shermanii JS* 4 × 10^9^ cfu vs. placebo	4 weeks	SCORAD at 4 weeks	All groups combined showed improvements in SCORAD: −65%.
							No group differences in changes in SCORAD (*p* = 0.27).
							In IgE sensitized infants; *LGG* group showed greater reductions in SCORAD (−26.1 points) when compared to placebo (−19.8 points) (*p* = 0.04).
Gruber et al., 2007	RCT	54/48	3–12 months	*LGG >* 5 × 10^9^ cfu vs. placebo	12 weeks	SCORAD A, B, and C Subscales at 12 weeks	Probiotic: A (−6.2 points); B (−1.2 points), C (−2.4 points)
							Placebo: A (−7.9 points), B (−1.6 points), C (−2.4 points)
							No significant group differences in changes in SCORAD subscales: (*p* = 0.60, 0.27, and 0.52, respectively)
Gore et al., 2012	RCT	35/36/40	3–6 months	*L. paracasei* 10^10^ cfu vs. *B. lactis* 10^10^ cfu vs. placebo	12 weeks	SCORAD at 12 weeks	*L. paracasei*: −51%
							*B. lactis*: −51%
							Placebo: −59%
							No significant differences in changes in SCORAD (*p* = 0.7).
Torii et al., 2011	RCT	26/24	4–15 years	*L. acidophilus 3* × 10^10^cfu vs. placebo	8 weeks	SMS at 8 weeks	Reductions in ADASI score were demonstrated in the probiotic group when compared to placebo (*p* < 0.05).
Farid et al., 2011	RCT	19/21	3 months–6 years	Synbiotic (prebiotic + *L. casei*, *L. rhamnosus*, *S. thermophilus*, *B. breve*, *L. acidophilus*, *B. infantis*, *L. bulgaricus*), 2 × 10^10^ cfu total dosage vs. placebo	8 weeks	SCORAD at 8 weeks	Synbiotic: −39 points
							Placebo: −20 points
							Significant group differences in changes in SCORAD (*p* = 0.005).
Gerasimov et al., 2010	RCT	43/47	12–36 months	*L. acidophilus* and *B. lactis* 10^10^cfu total dosage vs. placebo	8 weeks	SCORAD at 8 weeks	Probiotic: −14 points
							Placebo: −8 points
							Significant group differences in changes in SCORAD (*p* = 0.001).
							Significant correlation was seen between decreases in CD4 and CD25, and reductions in SCORAD values (*p* < 0.05).
Yesilova et al., 2012	RCT	20/19	1–13 years	*B. bifidum*, *L. acidophilus*, *L. casei*, *L. salivarius, 4* × 10^9^ cfu total dosage vs. placebo	8 weeks	SCORAD at 8 weeks	Probiotic: −65%
							Placebo: −46%
						Inflammatory markers	Significant group differences in changes in SCORAD (*p* = 0.002)
							Significant decreases in IL-6, IFN-γ, and IgE when compared to placebo (*p* < 0.01).
Prakoeswa et al., 2017	RCT	12/10	0–14 years old	*L. plantarum* 10^10^ cfu vs. placebo	12 weeks	SCORAD at 12 weeks	Probiotic: −37 points
							Placebo: −27 points
							Significant group differences in changes in SCORAD (*p* < 0.001).
Han et al., 2012	RCT	44/39	1–13 years	*L. plantarum* 10^10^cfu vs. placebo	12 weeks	SCORAD at 12 weeks	Probiotic: −9.1 pointes
							Placebo: −1.8 points
							Significant group differences in changes in SCORAD (*p* = 0.004).
							Total eosinophil count was decreased in the probiotic group (*p* = 0.023) as were IFN-γ (*p* < 0.001) and IL-4 (*p* = 0.049).
Yang et al., 2014	RCT	37/34	2–9 years old	*L. casei*, *L. plantarum*, *L. rhamnosus*, and *B. lactis 2* × 10^9^ cfu of each strain vs. placebo	6 weeks	EASI at 6 weeks	Probiotic: −35%
							Placebo: −46%
							No significant group differences in changes in EASI (*p* = 0.28)
Weston et al., 2005	RCT	26/27	6–18 months	*L. fermentum 2* × 10^9^ cfu vs. placebo	16 weeks	SCORAD at 16 weeks	Probiotic: −17 points (*p* = 0.03).
							Placebo: −12 points (*p* = 0.83)
							No data given on intergroup comparison
Wang et al., 2015	RCT	55/53/51/53	1–18 years	*L. paracasei 2* × 10^9^ cfu vs. *L. fermentum 2* × 10^9^ cfu vs. *L. paracasei* + *L. fermentum* 4 × 10^9^ cfu vs. placebo	12 weeks	SCORAD at 12 weeks	*L. paracasei*: −25 points
							*L. fermentum*: −24 points
							*L. paracasei + L. fermentum*: −28 points
						Inflammatory Markers	*Placebo*: −15 points
							Treatment groups demonstrated lower SCORAD index after treatment when compared to placebo (*p* < 0.001).
							IL-4 levels decreased after probiotic treatment (*p* = 0.04).
Wu et al., 2012	RCT	27/27	2–14 years	Synbiotic (*L. salivarius 4* × 10^9^ cfu with prebiotic) vs. prebiotic alone	8 weeks	SCORAD at 8 weeks	Synbiotic: 27 at week 8.
							Prebiotic: 36 at week 8
							Significant group differences in SCORAD at week 8 when controlling for baseline (*p* = 0.02)
Woo et al., 2010	RCT	41/34	2–10 years	*L. sakei* 10^10^ cfu vs placebo	12 weeks	SCORAD at 12 weeks	Probiotic: −31%
							Placebo: −13%
						Chemokine levels	Significant group differences in changes in SCORAD (*p* = 0.01).
							Treatment group demonstrated improvements in CCL17 and CCL27 (*p* = 0.03). Levels of cytokines were associated with SCORAD index (*p* < 0.001).
Sistek et al., 2006	RCT	25/24	1–10 years old	*L. rhamnosus* and *B. lactis 2* × 10^9^ cfu total dosage vs. placebo	12 weeks	SCORAD at 12 weeks	Ratio of probiotic to placebo treatment at treatment endpoint: 0.8 (p = 0.10).
							In food sensitized children, ratio of probiotic to placebo treatment at treatment endpoint: 0.73 (*p* = 0.047).
Navarro-Lopez et al., 2018	RCT	23/24	4–17 years old	*B. lactis*, *B. longum*, *L. casei*10^9^ cfu total dosage vs. placebo	12 weeks	SCORAD at 12 weeks	Probiotic: −83%
							Placebo: −24%
							Significant group difference in changes in SCORAD (*p* < 0.001)
Passeron et al., 2006	RCT	17/22	2–12 years old	Prebiotic + *L. rhamnosus* 1.2 × 10^9^ cfu vs. prebiotic alone	12 weeks	SCORAD at 12 weeks	Prebiotic + *L. rhamnosus*: −47%
							Prebiotic: −39%
							No group differences in changes in SCORAD (p = 0.54)
Folster-Holst et al., 2006	RCT	22/25	1–55 months	*LGG* 10^10^cfu vs. placebo	8 weeks	SCORAD at 8 weeks	Probiotic: −18%
							Placebo: −24%
							No significant group differences in changes in SCORAD.
Wu et al., 2017	RCT	30/32	4–48 months	*L. rhamnosus* 350 mg vs. placebo	8 weeks	SCORAD at 8 weeks	Probiotic: = −22 points
							Placebo: SCORAD = −12 points
							Significant group differences in SCORAD at week 8 when controlling for baseline (*p* = 0.01).
Shafiei et al., 2011	RCT	18/19	1–36 months	Synbiotic (7 strain probiotic 10^10^ cfu + prebiotic) vs. placebo	12 weeks	SCORAD at 12 weeks	All groups combined showed a significant decrease in SCORAD (−56%; *p* < 0.01).
							No group differences in changes in SCORAD (*p* > 0.05)

RCT, randomized controlled trial. SCORAD, severity assessment of atopic dermatitis. AD, atopic dermatitis. ADASI, atopic dermatitis area and severity index. SMS, symptom-medication score. EASI, eczema activity, and severity index. CFU, colony-forming unit.

**Table 2 children-06-00121-t002:** Common probiotics used in treating pediatric atopic dermatitis.

Common Probiotics Used for Pediatric Atopic Dermatitis
*L. acidophilus*
*L. plantarum*
*L. fermentum*
*L. rhamnosus*
*L. salivarius*
*L. sakei*
*L. casei*
*B. lactis*

**Table 3 children-06-00121-t003:** Vitamins and minerals for the treatment of pediatric atopic dermatitis.

Study	Study Type	Treated/Control (N)	Age Range	Vitamin/Mineral Treatment	Treatment Time	Outcome Measures	Outcomes
Januchowsi, 2009	RCT	21 patients total	6 months–18 years	Topical B12 vs. placebo	4 weeks	SCORAD at 4 weeks	B12: −4.5 points Placebo: −1.6 points Significant group difference in changes in SCORAD (*p* = 0.01)
Mabin et al., 1995	RCT	19/22	2–15 years	Oral pyridoxine hydrochloride vs. placebo	4 weeks	Skin severity score, daytime itch, nocturnal itch at 4 weeks	No significant differences between groups in any outcomes at the end of treatment. Skin severity score: *p* = 0.65 Daytime itch: *p* = 0.72 Nocturnal itch: *p* = 0.33
Ewing et al., 1991	RCT	25/25	1–16 years	Oral zinc sulfate vs. placebo	8 weeks	Surface area affected, degree of erythema, itch at 8 weeks	No change in disease severity across both groups.

RCT, randomized controlled trial. AD, atopic dermatitis.

**Table 4 children-06-00121-t004:** Chinese herbal treatments for pediatric atopic dermatitis.

Study	Study Type	Treated/Control (N)	Age Range	Chinese Herb Treatment	Treatment Time	Outcome Measures	Outcomes
Hon et al., 2007	RCT	42/43	5–21 years	*Flos ionicerae, Herba menthae, Cortex moutan, Rhizoma atractylodis, and Cortex phellodendri* (TCHM) vs. placebo	12 weeks	SCORAD at 12 weeks	TCHM: −15%
							Placebo: −18%
							No significant group differences in SCORAD
							TCHM group displayed a 1/3^rd^ reduction in the amount of topical corticosteroid used (*p* = 0.024).
Sheehan et al., 1992	Crossover Trial	37	1–18 years	*Ledebouriella seseloides, Potentilla chinensis, Anebia clematidis, Rehmannia glutinosa, Paeonia lactiflora, Lophatherum gracile, Dictamnus dasycarpus, Tribulus terrestris, Glycyrrhiza uralensis, and Schizonepeta tenuifolia* (TCHM) vs. placebo	8 weeks	Extent and severity of erythema and surface damage at 8 weeks	TCHM: 51% decrease in erythema scores and a 49% decrease in surface damage scores
							Placebo: 6.1% decrease in erythema scores and 6.2% decrease in surface damage scores.

RCT, randomized controlled trial. AD, atopic dermatitis. TCHM, traditional Chinese herbal medicine.

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
