# Peer review of "Complementary and Integrative Therapies for Childhood Atopic Dermatitis"

_children, 2019, doi:10.3390/children6110121_

Round 1

Reviewer 1 Report

This is a review article that used Pubmed database as a search engine to address whether complementary and integrative medicine would be helpful for the childhood atopic dermatitis. The authors found that certain strains of probiotics may be useful for the management of childhood atopic dermatitis. Please refer to the comments below.

How the pubmed search is conducted should be clarified. For example, what is year interval is defined in the search criteria? The result of the studies from each keyword combinations should be listed. The method section should be described in more details for consistency and reproducibility.   In abstract, the conclusion should be more specific than it is right now. The statement stated that certain strains …. (this should be clarified and specific). Table 1. The table should describe what the amount of each probiotics or probiotics was used in the study. The table 1 may be classified by the strains used. Outcome measurements should include the time of measurement.

Author Response

Thank you for your comments.  We have updated the Methods section to further delineate our approach. However, since this was not designed as a systematic review where all results of PubMed searches are tracked and each article found is documented, we do not have the exact numbers related to original search results.

Our Methods section was updated to state:

“A PubMed search included the terms “atopic dermatitis,” “children,” “pediatrics,” “integrative medicine,” “complementary medicine,” “alternative medicine,” “complementary,” “integrative,” “alternative,” “probiotic,” “herb,” “herbal medicine,” “vitamin,” and “relaxation.”  Results were filtered to only include human subjects. Dates included ranged from earliest entry to January of 2019.  Title, abstract, and full review were conducted by one reviewer.  We included only randomized, controlled trials (RCT) or crossover trials that examined participants less than or equal to eighteen years of age.  An exception was made for one study examining Chinese herbal medicine that incorporated individuals 5-21 years of age. 

 When using the above search terms, anywhere between approximately twenty to eighty trials resulted related to probiotics for the treatment of AD.  From those, ten were identified as meeting our study criteria, while the remaining ones cited below were found within the References section of additional articles. For the section related to vitamin supplementation, approximately ninety-five studies resulted from our search, with only one meeting our criteria.  An additional two studies were also found from the References section of articles.  For studies related to herbal medicine, approximately forty studies were identified on PubMed, two of which fit our classifications.”

We updated the abstract per your recommendation and added the specific strains that were found to be effective.  Specifically, we state:

Certain strains of probiotics, specifically L. plantarum and L. fermentum, may improve clinical severity scores in children with atopic dermatitis; however, additional trials are needed to more thoroughly delineate the effects of additional integrative therapies on childhood atopic dermatitis.”

Table 1 was updated to reflect the amount of each treatment used. It was also updated to include the time of measurement within the “Outcome Measure” column.

Reviewer 2 Report

Introduction, line 38. it's not easy to understand. I suggest including a comma before poor patient satisfaction.

3.1. Probiotics/prebiotics..., line 24. SCORAD indexes. It is not the most used form, better SCORAD index. Repeated several times in the manuscript.

3.1. Probiotics/prebiotics..., lines 38-39. The sentence "Interleukin (IL-4), interferon... compared to placebo." is written before the Clinical Trial explanation which this result belongs. I suggest writing it after the sentence in lines 39-41 in order to a better understanding.

3.1. Probiotics/prebiotics..., line 70. There are four trials with L. casei incorporated in the probiotic mix. You named 3 and after added another one, so that gives rise to error. 

3.1. Probiotics/prebiotics..., line 76. Navarro-Lopez et al. clinical trial used a probiotic mix composed by L. casei, B. lactis and B. longum

3.1. Probiotics/prebiotics..., line 80. There are eight trials with L. rhamnosus strain. 

3.1. Probiotics/prebiotics..., line 84. The age range does not match with the one in the table. Text: 2-12, table: 3-12.

3.1. Probiotics/prebiotics..., line 99. The SCORAD improving does not match with the one in the table. Text: 36%, table: 39%.

3.2. Vitamins/minerals for the..., line 114. There are two times and only one result: "SCORAD values at 2 and 4 weeks (-4.5 points)". 

3.3. Chinese herbs for the..., line 10. The patient number does not match with the one in the table. Text: 31, table: 37.

Author Response

Thank you for your helpful responses.  We have incorporated all of your requested changes. Inconsistencies were adjusted to reflect the accurate finding.

We also adjusted the Results to reflect that four trials incorporated L. casei as you noted.  Further, we incorporated the additional two trials referencing L. rhamnosus into the results section and altered our conclusions slightly to reflect the additional findings.

To this extent, in the Results section, we state: “Additionally, L. rhamnosus was combined with B. lactis and showed improvement in children who were food sensitized when compared to placebo (SCORAD ratio of probiotic to placebo treatment was 0.73 (p = 0.047) [12].  The strain was also effective when combined with a number of different probiotics and a prebiotic (-39 points) when compared to placebo (-20 points; p = 0.005) [17].”
